# Multimodal Deep Learning Model Unveils Behavioral Dynamics of V1 Activity in Freely Moving Mice

**Aiwen Xu**
Department of Computer Science
University of California, Santa Barbara
Santa Barbara, CA 93117
`aiwenxu@ucsb.edu`

**Yuchen Hou**
Department of Computer Science
University of California, Santa Barbara
Santa Barbara, CA 93117
`yuchenhou@ucsb.edu`

**Cristopher M. Niell**
Department of Biology, Institute of Neuroscience
University of Oregon
Eugene, OR 97403
`cniell@uoregon.edu`

**Michael Beyeler**
Department of Computer Science
Department of Psychological & Brain Sciences
University of California, Santa Barbara
Santa Barbara, CA 93117
`mbeyeler@ucsb.edu`

## Abstract

Despite their immense success as a model of macaque visual cortex, deep convolutional neural networks (CNNs) have struggled to predict activity in visual cortex of the mouse, which is thought to be strongly dependent on the animal's behavioral state. Furthermore, most computational models focus on predicting neural responses to static images presented under head fixation, which are dramatically different from the dynamic, continuous visual stimuli that arise during movement in the real world. Consequently, it is still unknown how natural visual input and different behavioral variables may integrate over time to generate responses in primary visual cortex (V1). To address this, we introduce a multimodal recurrent neural network that integrates gaze-contingent visual input with behavioral and temporal dynamics to explain V1 activity in freely moving mice. We show that the model achieves state-of-the-art predictions of V1 activity during free exploration and demonstrate the importance of each component in an extensive ablation study. Analyzing our model using maximally activating stimuli and saliency maps, we reveal new insights into cortical function, including the prevalence of mixed selectivity for behavioral variables in mouse V1. In summary, our model offers a comprehensive deep-learning framework for exploring the computational principles underlying V1 neurons in freely-moving animals engaged in natural behavior.

## 1 Introduction

Computational models have been crucial in providing insight into the underlying mechanisms by which neurons in the visual cortex respond to external stimuli. Deep convolutional neural networks (CNNs) have had immense success as predictive models of the primate ventral stream, in cases where

37th Conference on Neural Information Processing Systems (NeurIPS 2023).

the animal was passively viewing stimuli or simply maintaining fixation [1–5]. Despite their success, these CNNs are poor predictors of neural responses in mouse visual cortex [6], which is thought to be shallower and more parallel than that of primates [7, 8]. According to the best models in the literature [9–14], the mouse visual system is more broadly tuned and operates on relatively low-resolution inputs to support a variety of behaviors [15]. However, these models were limited to predicting neural responses to controlled (and potentially ethologically irrelevant) stimuli that were passively viewed by head-fixed animals.

Movement is a critical element of natural behavior. In the visual system, eye and head movements during locomotion and orienting transform the visual scene in potentially both beneficial (e.g., by providing additional visual cues) and detrimental ways (e.g., by introducing confounds due to self-movement) [16–18]. Movement-related activity is widespread in mouse cortex [19, 20] and prevalent in primary visual cortex (V1) [21, 22]. For instance, V1 neurons of freely moving mice show robust responses to head and eye position [23, 24], which may contribute a multiplicative gain to the visual response [25] that cannot be replicated under head fixation. V1 activity may be further modulated by variables that depend on the state of the animal and its behavioral goals [20, 22, 26, 27]. However, how these behavioral variables may integrate to modulate visual responses in V1 is unknown. Furthermore, a comprehensive predictive model of V1 activity in freely moving animals is still lacking.

To address these challenges, we make the following contributions:

- We introduce a multimodal recurrent neural network that integrates gaze-contingent visual input with behavioral and temporal dynamics to explain V1 activity during natural vision in freely moving mice.
- We show that the model achieves state-of-the-art predictions of V1 activity during free exploration based on visual input and behavior, demonstrating the ability to accurately model neural responses in the dynamic regime of movement through the visual scene.
- We uncover new insights into cortical neural coding by analyzing our model with maximally activating stimuli and saliency maps, and demonstrate that mixed selectivity of visual and behavioral variables is prevalent in mouse V1.

## 2 Related Work

The mouse, as a model organism, offers unparalleled experimental access to the mammalian cerebral cortex [28]. Computational models of mouse V1, including generalized linear models (GLMs) [25, 29] and customized models mimicking the mouse visual hierarchy [30], have been crucial in providing deeper insights into the range of computations performed by visual cortex. More recently, deep CNNs have also been used to model mouse V1 [9–13, 31–34].

Despite their success in predicting neural activity in the macaque visual cortex [35], deep CNNs trained on ImageNet have had limited success in predicting mouse visual cortical activity [6]. This is perhaps not surprising, as most ImageNet stimuli belong to static images of human-relevant semantic categories and may thus be of low ethological relevance for rodents. More importantly, these deep CNNs may not be the ideal architecture to model mouse visual cortex, which is known to be shallower and more parallel than primate visual cortex [36, 37]. In addition, mice are known to have lower visual acuity than that of primates [7, 8], and much of their visual processing may be devoted to active, movement-based behavior rather than passive analysis of the visual scene [22, 38, 39]. Although the majority of V1 neurons is believed to encode low-level visual features [40], their activity is often strongly modulated by behavioral variables related to eye and head position [23–25], locomotion [18, 21, 22], arousal [27, 41], and the recent history of the animal [26]. Furthermore, mouse V1 is highly interconnected with both cortical and subcortical brain areas, which contrasts with feedforward, hierarchical models of visual processing [22].

A common architectural approach that has proved quite successful is to split the network into different components (first introduced by [11]):

- a "core" network, which typically consist of a CNN used to extract convolutional features from the visual stimulus [11, 12, 25, 42], sometimes in combination with a recurrent network [11];
- a "shifter" network, which mimics gaze shifts by learning a (typically affine) transformation from head- to eye-centered coordinates, either applied to the pixel input [11, 25] or a CNN layer [12];
- a "readout" network, which learns a mapping from artificial to biological neurons [11, 12, 42].

Owing to the difficulty of developing a predictive model of mouse cortex, Willeke *et al.* [14] recently invited submissions to the Sensorium competition held at NeurIPS '22. The competition introduced a benchmark dataset of V1 neural activity recorded from head-fixed mice on a treadmill viewing static images, with simultaneous measurements of running speed, pupil size, and eye position. A baseline model was provided as well, which consisted of a 4-layer CNN core in combination with a shifter and readout network [12]. Even though 26 teams submitted 194 different models, the overall improvement to the baseline performance was modest, raising the single trial correlation from .287 to .325 in the Sensorium and from .384 to .453 in the Sensorium+ competition. Architectural innovations (e.g., Transformers, Normalizing Flows, YOLO, and knowledge distillation), were unable to make an impact, as most improvements were gained from ensemble methods. A promising direction was taken by the winning model, which attempted to learn a latent representation of the "brain state" from the various behavioral variables, inspired by [20]. However, the model utilized the timestamps of the test set to estimate recent neuronal activities, which the other competitors did not have access to.

Taken together, we identified three main limitations of previous work that this study aims to address:

- **Head-fixed preparations.** Most previous models operated on data from animals in head-fixed conditions with static stimuli, which do not mirror natural behavior and thus provide limited insight into visual processing in real-world environments. In contrast, the present work is applied to state-of-the-art neurophysiological recordings of V1 activity in freely moving mice. This represents a dramatic shift in the "parameter space" of visual input, from static images to dynamic, real-world visual input. One could imagine that this will make the modeling process more difficult, because the stimulus set is more complex, or easier, because it is more matched to the computational challenge the brain evolved for.
- **Limited influence of behavioral state.** Previous models often limited the influence of behavioral state to eye measurements and treadmill running speed, which were either concatenated with the visual features [14, 41], utilized in the shifter network to determine the gaze-contingent retinal input [11, 14], or used to predict a multiplicative gain factor [11].
- **Missing temporal dynamics.** Most previous modeling works ignored the temporal factors that might influence V1 activity and overlooked the dynamic nature of visual processing (but see [11]). We overcome this limitation by utilizing approximately 1-hour-long recordings of three mice freely exploring an arena, and our model is capable of handling continuous data streams of any length.

## 3 Methods

**Head-mounted recording system** We had access to data from three adult mice who were freely exploring $48\,\mathrm{cm}$ long $\times$ $37\,\mathrm{cm}$ wide $\times$ $30\,\mathrm{cm}$ high arena (Fig. 1A), collected with a state-of-the-art recording system [25] that combined high-density silicon probes with miniature head-mounted cameras (Fig. 1B). One camera was aimed outwards to capture the visual scene from the mouse's perspective ("worldcam") at $16\,\mathrm{ms}$ per frame (downsampled to $60 \times 80$ pixels). A second camera, aimed at the eye, was used to extract eye position ($\theta$, $\phi$) and pupil radius ($\sigma$) at $30\,\mathrm{Hz}$ using DeepLabCut [43]. Pitch ($\rho$) and roll ($\omega$) of the mouse's head position were extracted at $30\,\mathrm{kHz}$ from the inertial measurement unit (IMU). $\theta$, $\phi$, $\rho$, and $\omega$ allowed for the worldcam video to be corrected for eye movements: A 3-layer fully-connected shifter network (where each linear layer was accompanied by Tanh and BatchNorm) was trained to predict a rotation (bounded by $\pm 36°$) and a shift (bounded by $\pm 16$ pixels horizontally and $\pm 12$ pixels vertically) based on $\theta$, $\phi$, $\rho$, and $\omega$ to convert each frame to head- to eye-centered coordinates. Locomotion speed ($s$) was estimated from the top-down camera feed using DeepLabCut [43]. Electrophysiology data was acquired at $30\,\mathrm{kHz}$ using a $11\,\mu\mathrm{m} \times 15\,\mu\mathrm{m}$ multi-shank linear silicon probe (128 channels) implanted in the center of the left monocular V1, then bandpass-filtered between $0.01\,\mathrm{Hz}$ and $7.5\,\mathrm{kHz}$, and spike-sorted with Kilosort 2.5 [44]. Single units were selected using Phy2 (see [45]) and inactive units (mean firing rate $< 3\,\mathrm{Hz}$) were removed. This yielded 68, 32, and 49 active units for Mouse 1–3, respectively. To prepare the data for machine learning, all data streams were deinterlaced and resampled at $20.83\,\mathrm{Hz}$ ($48\,\mathrm{ms}$ per frame; Fig. 1C). For a more detailed description of the dataset, see Appendix A and Ref. [25].

**Model architecture** We used a 3-layer CNN (kernel size 7, $128 \times 64 \times 32$ channels) to encode the visual stimulus. Each convolutional layer was followed by a BatchNorm layer, a ReLU, and a Dropout layer (0.5 rate). A fully-connected layer transformed the learned visual features into a

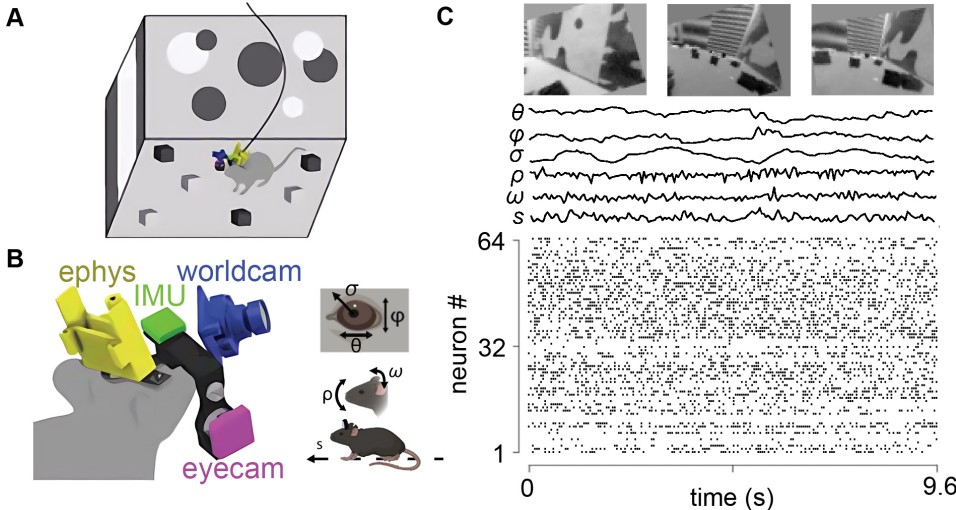

Figure 1: Schematic of the head-mounted recording system for freely moving mice (adapted from [25]). A) Three mice freely explored a $48\,\mathrm{cm}$ long $\times$ $37\,\mathrm{cm}$ wide $\times$ $30\,\mathrm{cm}$ high arena. B) Preparation included a silicon probe for electrophysiological recording in V1 (yellow), miniature cameras for recording the mouse's eye position and pupil size ($\theta$, $\phi$, and $\sigma$; magenta), and visual scene (blue), and inertial measurement unit for measuring head orientation ($\rho$ and $\omega$; green). C) Sample data from a $9.6\,\mathrm{s}$ period during free movement showing (from top) visual scene, horizontal and vertical eye position, pupil size, head pitch and roll, locomotor speed, and a raster plot of 64 units.

visual feature vector, $\boldsymbol{v}$ (Fig. 2, *top-right*). In a purely visual version of the model, $\boldsymbol{v}$ was fed into a fully-connected layer, followed by a softplus layer, to yield a neuronal response prediction.

To encode behavioral state, we constructed an input vector from different sets of behavioral variables:

- $\mathcal{S}$: all behavioral variables used in the Sensorium+ competition [14], consisting of running speed ($s$), pupil size ($\sigma$), and its temporal derivative ($\dot{\sigma}$);
- $\mathcal{B}$: all behavioral variables used in [25], consisting of eye position ($\theta$, $\phi$), head position ($\rho$, $\omega$), pupil size ($\sigma$), and running speed ($s$);
- $\mathcal{D}$: the first-order derivatives of the variables in $\mathcal{B}$, namely $\dot{\theta}$, $\dot{\phi}$, $\dot{\omega}$, $\dot{\rho}$, $\dot{\sigma}$, and $s$.

To test for interactions between behavioral variables, these sets could also include the pairwise multiplication of their elements; e.g., $\mathcal{B}_{\times} = \{b_i b_j \ \forall \ (b_i, b_j) \in \mathcal{B}\}$. The input vector was then passed through a batch normalization layer and a fully connected layer (subjected to a strong L1 norm for feature selection) to produce a behavioral vector, $\boldsymbol{b}$.

We then concatenated the vectors $\boldsymbol{v}$, $\boldsymbol{b}$, and their element-wise product $\boldsymbol{v} \odot \boldsymbol{b}$ (all calculated for each individual input frame), fed them through a batch normalization layer, and input them to a 1-layer gated recurrent unit (GRU) (hidden size of 512). To incorporate temporal dynamics, we constructed different versions (GRU$_k$) of the model that had access to $k$ previous frames. A fully-connected layer and a softplus activation function were applied to yield the neuronal response prediction.

**Training and model evaluation** Since the visual input depended on the movement of the mouse and the mouse could be in very different behavioral states over the length of the recording, the data was highly inhomogeneous across time. To deal with the continuous and dynamic nature of the data, we therefore split the $\sim 1\,\mathrm{h}$-long recording into 10 consecutive segments. The first $70\,\%$ of each segment were then reserved for training (including an 80-20 validation split) and the remaining $30\,\%$ for testing.

Note that it is unlikely for data to "leak" from the train segment into the test segment. While it is possible that the mouse could have been exploring the same part of the arena at different segments of the recording, it was free to move its head, eyes, and body as it saw fit. Thus two duplicate data points could only be produced by the animal exactly duplicating the time courses of its eye, head, and body movement in the exact same location of the arena.

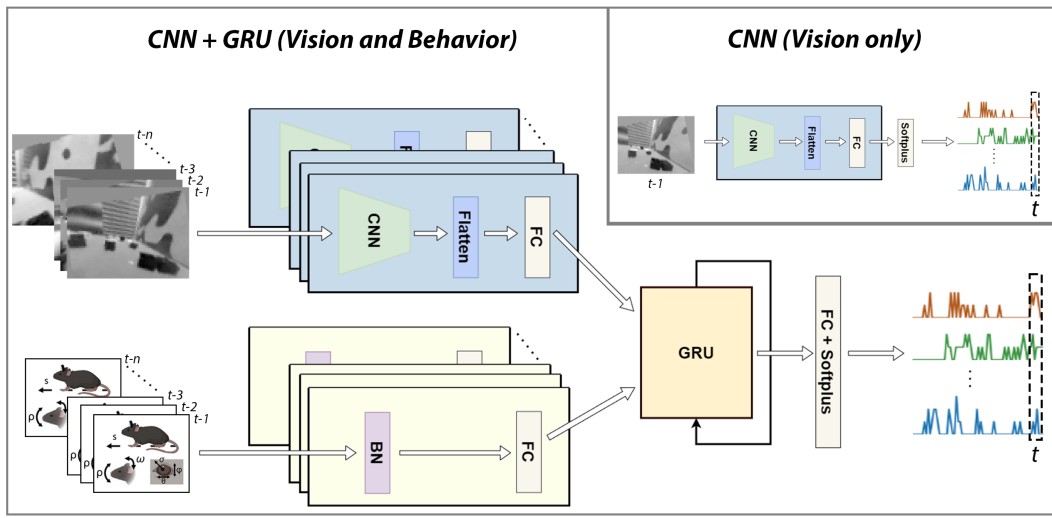

Figure 2: Model architecture diagram. The vision-only network (top-right) was a CNN network, predicting the neural activity at time $t$ given the visual input at time $t-1$ (48 ms bins). The full model combined the CNN with a behavioral encoder and a gated recurrent unit (GRU), predicting the neural activity at time $t$ given the visual and behavioral inputs from time $t-1$ to $t-n$.

Models were separately trained on the data from each mouse. Model parameters were optimized with Adam (batch size: 256, CNN learning rate: .0001, full model: .0002) to minimize the Poisson loss between predicted neuronal response ($\hat{r}$) and ground truth ($r$) : $\frac{1}{N}\sum_{i=1}^{N}(\hat{r}_i - r_i \log \hat{r}_i)$, where $N$ denotes the number of recorded neurons for each mouse. We used early stopping on the validation set (patience: 5 epochs), which led all models to converge in less than 50 epochs. Due to the large number of hyper-parameters, the specific network and training settings were determined using a combination of grid search and manual exploration on a validation set (see Appendix B).

To evaluate model performance, we calculated the cross-correlation ($cc$) between a smoothed version (2 s boxcar filter) of the predicted and ground-truth response for each recorded neuron [25].

All models were implemented in PyTorch and trained on an NVIDIA RTX 3090 with 24GB memory. All code, data used to train the models, and weights of the trained model can be found at `https://github.com/bionicvisionlab/2023-Xu-Multimodal-Mouse-V1`.

**Maximally activating stimuli** We used gradient ascent [42] to discover the visual stimuli that most strongly activate a particular model neuron in our network. The visual input was initialized with noise sampled in $\mathcal{N}(.5, 2)$. The behavioral variables were initialized to a vector of all ones, and updated in the loop with the visual stimuli. We used the Adam optimizer to repeatedly add the gradient of the target neuron's activity with respect to its inputs. We also applied L2 regularization (weight of .02) and Laplacian regularization (weight of 0.01) [46] on the image. This procedure was repeated 6400 times. The resulting, maximally activating visual stimuli were smoothed with a Butterworth filter (low-pass, .05 cutoff frequency ratio) to reduce the impact of high-frequency noise.

**Saliency map** We computed a saliency map [47] of the behavioral vector for each neuron to discover which behavioral variables contributed most strongly to each model neuron's activity. We iterated through the test dataset, recorded the gradient of each behavioral input with respect to each neuron's prediction, and then averaged the gradients per neuron to obtain the saliency map.

## 4 Results

**Mouse V1 activity is best predicted with a 3-layer CNN** To determine the purely visual contribution to V1 responses, we experimented with a large number of vision architectures (see Appendix B). In the end, a vanilla 3-layer CNN (kernel size 7, $128 \times 64 \times 32$ channels) yielded

the best cross-correlation between predicted and ground-truth responses (Table 1), outperforming the best autoencoder architecture (kernel size: 7, encoder: $64 \times 128 \times 256$ channels, decoder: $256 \times 128 \times 64$ channels), ResNet-18 [48] (a 20-layer CNN with the first input channel being replaced by 1), EfficientNet-B0 [49] (a 65-layer CNN with the first input channel being replaced by 1), and the Sensorium baseline [12] (a 4-layer CNN with a readout network). The greatest improvement in cross-correlation was achieved for Mouse 1.

**Behavioral variables improve most neuronal predictions** Once we identified the 3-layer CNN as the best visual encoder, we added the different sets of behavioral variables to the network. To allow for a fair comparison with the Sensorium+ baseline [14], we first limited ourselves to $\mathcal{S} = \{\sigma, \dot{\sigma}, s\}$, but then gradually added more behavioral variables ($\mathcal{B}$) [25] as well the derivatives of these variables ($\mathcal{D}$) and multiplicative pairs ($\mathcal{B}_\times$ and $\{\mathcal{B} \cup \mathcal{D}\}_\times$).

The results are shown in Table 2. All models were able to outperform the Sensorium+ baseline, and the addition of behavioral variables and their interactions further improved model performance. Note that although the full model used a GRU to combine visual and behavioral features, the input sequence length was always 1 (i.e., $\mathrm{GRU}_1$). That being said, it is possible that the GRU learned long-term correlations that the Sensorium+ baseline model did not have access to. Nevertheless, the biggest performance improvements were gained through the addition of behavioral variables related to head and eye position (which are present in $\mathcal{B}$ but not in $\mathcal{S}$), their derivatives ($\mathcal{D}$), and multiplicative interactions between these variables ($\{\mathcal{B} \cup \mathcal{D}\}_\times$).

We also wondered whether the prediction of only some V1 neurons would benefit from the addition of these behavioral variables. To our surprise, the cross-correlation between predicted and ground-truth responses improved for almost all recorded V1 neurons (Fig. 3).

| Model | Mouse 1 $cc \uparrow$ | Mouse 1 MSE $\downarrow$ | Mouse 2 $cc \uparrow$ | Mouse 2 MSE $\downarrow$ | Mouse 3 $cc \uparrow$ | Mouse 3 MSE $\downarrow$ |
|---|---|---|---|---|---|---|
| CNN | **.596 ± .134** | **.0626** | **.424 ± .141** | **.100** | **.552 ± .138** | **.0908** |
| Autoencoder | .555 ± .140 | .0710 | .370 ± .145 | .112 | .521 ± .144 | .0974 |
| ResNet-18 [48] | .517 ± .159 | .0782 | .366 ± .171 | .107 | .511 ± .138 | .0944 |
| EfficientNet-B0 [49] | .542 ± .153 | .0694 | .393 ± .165 | .103 | .510 ± .127 | .0965 |
| Sensorium [12] | .519 ± .149 | .0754 | .381 ± .128 | .119 | .497 ± .136 | .100 |

Table 1: Best-performing vision models, compared to the Sensorium baseline [12] (see Appendix B for more). Best-performing networks are indicated in bold. $cc$: cross-correlation, mean $\pm$ standard deviation across neurons ($\uparrow$: the higher the better), MSE: mean-squared error ($\downarrow$: the lower the better).

| | Feature Set | Mouse 1 $cc \uparrow$ | Mouse 1 MSE $\downarrow$ | Mouse 2 $cc \uparrow$ | Mouse 2 MSE $\downarrow$ | Mouse 3 $cc \uparrow$ | Mouse 3 MSE $\downarrow$ |
|---|---|---|---|---|---|---|---|
| CNN+GRU$_1$ | $\{\mathcal{B} \cup \mathcal{D}\}_\times$ | **.646 ± .136** | **.0543** | **.508 ± .166** | **.0917** | **.607 ± .132** | .0801 |
| | $\mathcal{B} \cup \mathcal{D}$ | .644 ± .141 | **.0543** | .467 ± .180 | .0966 | .599 ± .134 | **.0797** |
| | $\mathcal{B}_\times$ | .639 ± .141 | .0555 | .484 ± .165 | .0925 | .593 ± .132 | .0812 |
| | $\mathcal{B}$ | .641 ± .136 | .0555 | .450 ± .158 | .0962 | .583 ± .136 | .0825 |
| | $\mathcal{S}$ | .623 ± .142 | .0551 | .498 ± .154 | .0911 | .579 ± .135 | .0828 |
| | Sensorium+ | .540 ± .138 | .0696 | .441 ± .181 | .101 | .487 ± .146 | .0975 |

Table 2: CNN+GRU$_1$ model, compared to the Sensorium+ baseline [12], trained on different behavioral feature sets. $\mathcal{S} = \{\sigma, \dot{\sigma}, s\}$: the set of variables used in the Sensorium+ competition [14]. $\mathcal{B} = \{\theta, \phi, \omega, \rho, \sigma, s\}$: the set of variables from [25]. $\mathcal{D} = \{\dot{\theta}, \dot{\phi}, \dot{\omega}, \dot{\rho}, \dot{\sigma}, s\}$: the derivatives of $\mathcal{B}$. $\mathcal{A}_\times = \{a_i a_j \; \forall \; (a_i, a_j) \in \mathcal{A}\}$ denotes the set of all multiplicative pairs. $\cup$ denotes the union operator. Best performing networks are indicated in bold. $cc$: cross-correlation, mean $\pm$ standard deviation across neurons ($\uparrow$: the higher the better), MSE: mean-squared error ($\downarrow$: the lower the better).

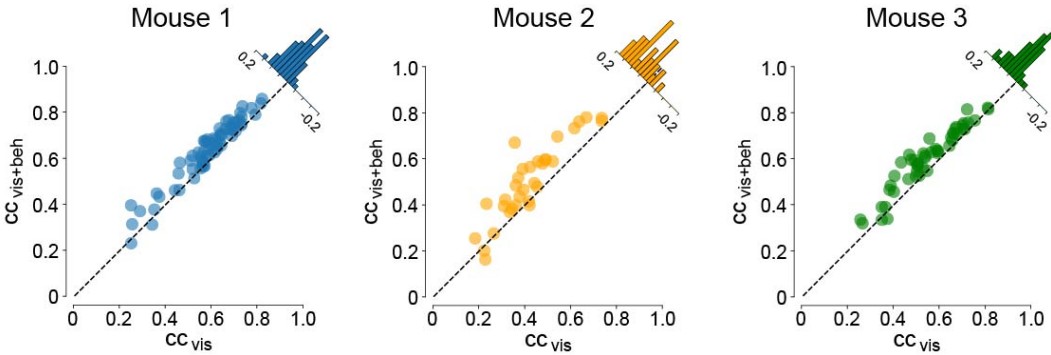

Figure 3: The integration of behavioral variables improved the cross-correlation ($cc$) for the majority of neurons. Each dot represents a neuron. A dot above the dashed diagonal indicates a higher $cc$ with the inclusion of behavioral variables. Histograms (small insets) illustrate the distribution of the improvement in $cc$ across the neuronal population.

**Access to longer series of data in time further improves predictive performance**  After we identified the full behavioral feature set ($\{\mathcal{B} \cup \mathcal{D}\}_\times$) as the one yielding the best model performance, we extended the GRU's temporal dependence by allowing the input to vary from one frame ($48\,\mathrm{ms}$) to a total of eight frames ($384\,\mathrm{ms}$), and assessed the model's performance.

The results are shown in Table 3. The amount of temporal information needed by the model to reach peak predictive performance was similar across mice ($288\,\mathrm{ms}$, $192\,\mathrm{ms}$, and $192\,\mathrm{ms}$ in terms of cross-correlation, $192\,\mathrm{ms}$, $192\,\mathrm{ms}$, and $192\,\mathrm{ms}$ in terms of mean-squared error, respectively). This indicates that temporal information is important for predicting dynamic neural activity. However, the dependence on temporal information has a limit, and different neurons in V1 might possess different temporal capacities.

**Well-defined visual receptive fields emerge**  To assess whether the CNN+GRU$_1$ model learned meaningful visual receptive fields, we used gradient ascent (see Methods) to find the maximally activating stimulus for each neuron. Receptive fields for the 32 best-predicted neurons are shown in Fig. 4. Interestingly, most of them had well-defined excitatory and inhibitory subregions, often resembling receptive fields of orientation-selective neurons. Most excitatory and inhibitory subregions spanned approximately $30°$ of visual angle (the full width of the frame, 80 pixels, roughly corresponding to $120°$ of visual angle), which is roughly on the same order of magnitude compared to receptive field sizes typically observed in mouse V1, varying from $10°$ to $30°$ [8, 25, 50].

Receptive fields were noticeably different across mice. Whereas Mouse 1 and 3 had visual receptive fields with strongly excitatory subregions, most model neurons for Mouse 2 appeared to be inhibited

|  |  | Mouse 1 | | Mouse 2 | | Mouse 3 | |
| --- | --- | ---: | ---: | ---: | ---: | ---: | ---: |
| Model | History | $cc\uparrow$ | MSE$\downarrow$ | $cc\uparrow$ | MSE$\downarrow$ | $cc\uparrow$ | MSE$\downarrow$ |
| CNN+GRU$_1$ | $48\,\mathrm{ms}$ | $.646 \pm .136$ | $.0543$ | $.508 \pm .166$ | $.0917$ | $.607 \pm .132$ | $.0801$ |
| CNN+GRU$_2$ | $96\,\mathrm{ms}$ | $.649 \pm .139$ | $.0528$ | $.506 \pm .174$ | $.0898$ | $.607 \pm .133$ | $.0811$ |
| CNN+GRU$_3$ | $144\,\mathrm{ms}$ | $.653 \pm .139$ | $.0528$ | $.528 \pm .160$ | $.0843$ | $.604 \pm .134$ | $.0790$ |
| CNN+GRU$_4$ | $192\,\mathrm{ms}$ | $.650 \pm .142$ | $\mathbf{.0525}$ | $\mathbf{.566 \pm .169}$ | $\mathbf{.0799}$ | $\mathbf{.614 \pm .136}$ | $\mathbf{.0773}$ |
| CNN+GRU$_5$ | $240\,\mathrm{ms}$ | $.645 \pm .144$ | $.0556$ | $.519 \pm .177$ | $.0933$ | $.598 \pm .134$ | $.0807$ |
| CNN+GRU$_6$ | $288\,\mathrm{ms}$ | $\mathbf{.654 \pm .142}$ | $.0526$ | $.549 \pm .175$ | $.0823$ | $.598 \pm .138$ | $.0798$ |
| CNN+GRU$_7$ | $336\,\mathrm{ms}$ | $.644 \pm .148$ | $.0534$ | $.533 \pm .169$ | $.0931$ | $.596 \pm .133$ | $.0806$ |
| CNN+GRU$_8$ | $384\,\mathrm{ms}$ | $.646 \pm .146$ | $.0547$ | $.546 \pm .179$ | $.0840$ | $.594 \pm .141$ | $.0825$ |

Table 3: CNN+GRU$_k$ model trained with input from $k$ timesteps on the full feature set ($\{\mathcal{B} \cup \mathcal{D}\}_\times$). Best performing networks are indicated in bold. $cc$: cross-correlation, mean $\pm$ standard deviation ($\uparrow$: the higher the better), MSE: mean-squared error ($\downarrow$: the lower the better).

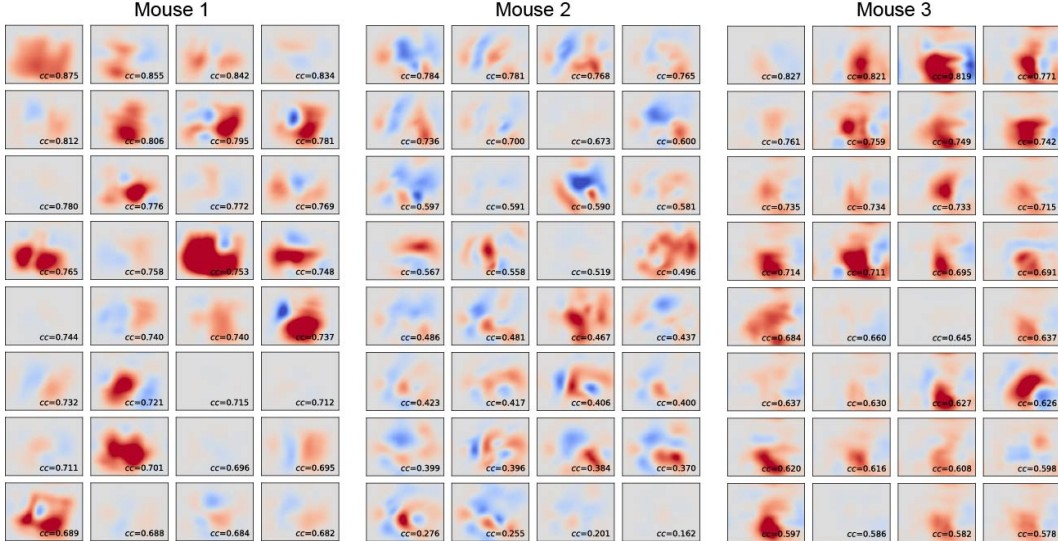

Figure 4: The maximally activating stimuli learned in CNN+GRU$_1$, generated via gradient ascent. The 32 neurons with the highest cross-correlation ($cc$) from each mouse are shown, sorted by $cc$.

by visual signals (same colorbar across panels). In addition, several model neurons lacked pronounced visual receptive fields, indicating that they were more strongly driven by behavioral variables. Even though model fits were repeated with different initial values for the behavioral variables, the resulting visual receptive fields looked qualitatively the same (see Appendix C), thus demonstrating the validity of the generated receptive fields. In addition, even some of the best-predicted neurons lack a pronounced or spatially structured receptive field, implying that these neurons could be primarily driven by behavioral variables.

**Analysis of behavioral saliency maps reveals different types of neurons**   Intrigued by the fact that some neurons lacked pronounced visual receptive fields, we aimed to analyze the influence of behavioral state on the predicted neuronal response by performing a saliency map analysis on the behavioral inputs (see Methods). Since different behavioral variables operate on different input ranges, we first standardized the saliency map activities for each behavioral variable across the model neuron population. Saliency map activities further than 1 standard deviation from the mean were then interpreted as "driving" the neuron, allowing us to categorize each neuron as being driven by one or multiple behavioral variables (Fig. 5).

We first asked which neurons in our model were driven by which behavioral variables (Fig. 5, *top*). Consistent with [25], we found a large fraction of model neurons driven by eye and head position, and smaller fractions driven by locomotion speed and pupil size. Approximately 20-30% of neurons were not driven by any of these behavioral variables, rendering their responses purely visual.

However, a particular neuron could be driven by multiple behavioral variables. Repeating the above analysis, we found that most model neurons showed mixed selectivity (i.e., responding to different categories of information, such as visual and motor signals, or stimulus and reward signals), with only a minority of cells responding exclusively to a single behavioral variable, (Fig. 5, *middle*). Adding the interaction terms between behavioral variables (Fig. 5, *bottom*) did not change the fact that most model V1 neurons encoded combinations of multiple behavioral variables, often relating information about the animal's eye position to head position and locomotor speed.

## 5   Discussion

In this paper, we propose a deep recurrent neural network that achieves state-of-the-art predictions of V1 activity in freely moving mice. We discovered that our model outperforms previous models under these more naturalistic conditions, which could be attributed to the better alignment of this data with the computations performed by the mouse visual system, based on its natural visual environment and

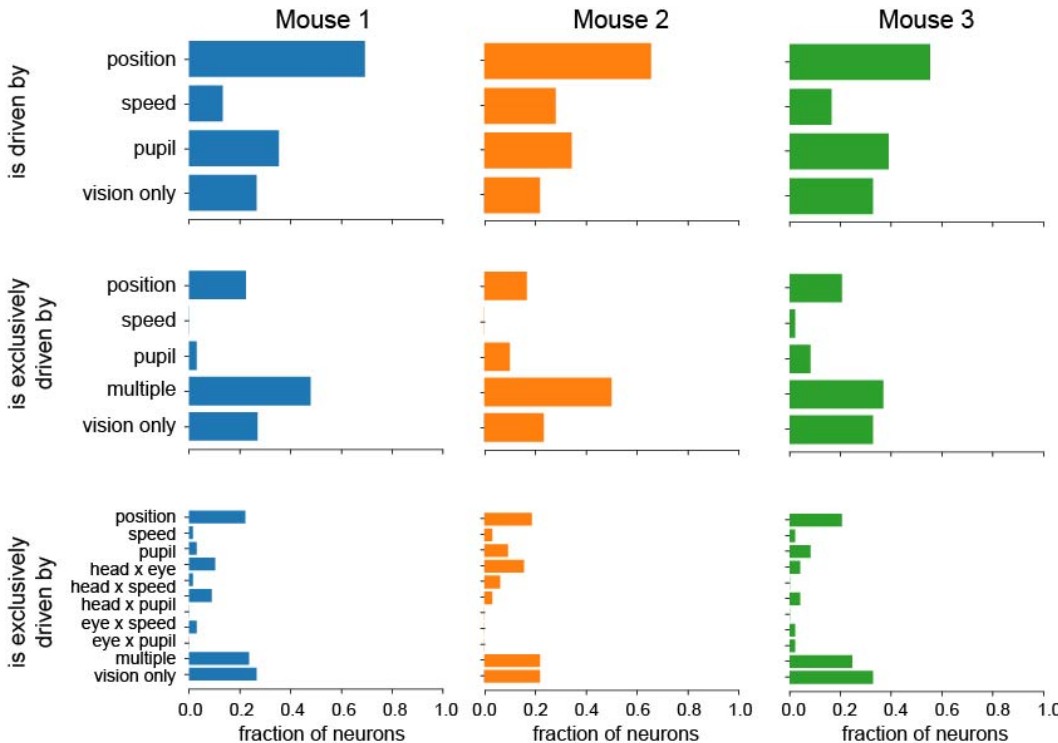

Figure 5: Effect of behavioral variables on model neuron activity, inferred by the saliency analysis. A) Fraction of neurons that are "driven by" (i.e., their saliency map activation is further than 1 standard deviation from the mean) different behavioral variables (similar to [25]). A neuron that responds to (e.g.) both position and speed may be counted twice. Neurons without a strong behavioral drive are categorized as "vision only". B) Fraction of neurons that are uniquely driven by a specific behavioral variable. Still, a large fraction of neurons are driven by multiple behavioral variables. C) Same as B), but split with interaction terms.

behavioral characteristics. Similar to previous models, we found that a simple CNN architecture is sufficient to predict the visual response properties of cells in mouse V1.

In addition, mouse V1 is known to be strongly modulated by signals related to the movement of the animal's eyes, head, and body [21, 22, 25], which are severely restricted in head-fixed preparations. Models trained on head-fixed preparations may thus be limited in their predictive power. In contrast, our model was able to predict V1 activity on a 1-hour continuous data stream, during which the animal freely explored a real-world arena. Our analyses demonstrate the impact of the animal's behavioral state on V1 activity and reveal that most model V1 neurons exhibit mixed selectivity to multiple behavioral variables.

**Accurate predictions of mouse V1 activity under natural conditions**   Our brains did not evolve to view stationary stimuli on a computer screen. However, most research on neural coding in vision has been conducted under head-fixed conditions, which do not mirror natural behavior and thus provide limited insight into visual processing in real-world environments. Some visual functions mediated by the ventral stream, such as identifying faces and objects, resemble this condition, but the real visual environment is constantly shifting due to self-motion, leading to dynamic activities such as navigation or object reaching, typically mediated by the dorsal stream. To truly understand visual perception in natural environments, we need to capture the computational principles when the subjects are in motion.

In this research, we take the initial steps towards this by modeling a novel data type encompassing neural activity coupled with a visual scene captured from a freely moving animal's perspective. This represents a dramatic (but, in our opinion, crucial) shift in the "parameter space" of visual input, from static images projected on a screen to dynamic, real-world visual input.

Surprisingly, visual responses were best predicted with a standard three-layer ("vanilla") CNN (Table 1), as compared to a multitude of more sophisticated models that included autoencoders, variational autoencoders, filter bank models, and pre-trained ResNet and EfficientNet architectures. One possible explanation might be that the neurons in our dataset were selective for other behavioral inputs that we did not have access to, and that the vanilla CNN architecture imposed the fewest assumptions about how visual input contributed to the neural activity. In addition, visual receptive fields for Mouse 2 were noticeably different from the other two mice, exhibiting pronounced inhibitory subregions (Fig. 4, *center*). This is consistent with the fact that the cortical probes of Mouse 2 were more superficial compared to the other two mice [25], so the recorded neurons may have both different anatomical inputs and different visual responses.

**Mixed selectivity of behavioral variables**    Our experiments demonstrated that the models incorporating behavioral variables and their interactions performed substantially better than the models relying exclusively on visual inputs. Moreover, our saliency map analysis showed that only around 25% of model neurons could be considered purely visual, with the majority of model neurons driven by multiple behavioral variables.

This widespread mixed selectivity is consistent with previous literature suggesting that V1 neurons may be modulated by a high-dimensional latent representation of several behavioral variables related to the animal's movement, recent experiences, and behavioral goals [20]. It is also consistent with the idea of a basis function representation [51, 52], which allows a population of neurons to conjunctively represent multiple behaviorally relevant variables. Such representations are often employed by higher-order visual areas in primate cortex to implement sensorimotor transformations [17, 51, 53]. It is intriguing to find computational evidence for such a representation as early as V1 in the mouse. Future computational studies should therefore aim to study the mechanisms by which V1 neurons might construct a nonlinear combination of behavioral signals.

**Limitations and future work.**    While our study opens a new perspective on modeling neural activity during natural conditions, there are a few limitations that need to be acknowledged. First, our data was relatively limited (around 50 neurons per animal, for 3 animals). The development of a Sensorium-style standardized dataset [14] for freely-moving mice would significantly benefit future research in this area, enabling more robust comparisons between different modeling approaches. Second, it would be beneficial to integrate other modalities that are known to be encoded in mouse V1 into the model. One such example is reward signals [54], which could provide additional information about the animal's decision-making processes and motivations during exploration.

## Acknowledgments

This work was supported by the National Institute of Neurological Disorders and Stroke of the National Institutes of Health under Award Number R01-NS121919. The content is solely the responsibility of the authors and does not necessarily represent the official views of the National Institutes of Health.

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

# Appendix

## A    Data Acquisition

The full procedure of data collection and data preprocessing is described in detail in Ref. [25], but are briefly described below for the reader's convenience.

Visual scenes from the mouse's perspective, neural activity in V1, and various behavioral variables were simultaneously recorded from three adult mice who were freely exploring an arena with a state-of-the-art head-mounted recording system [25]. The recording system consisted of high-density silicon probes, two miniature cameras and an inertial measurement unit (IMU). Electrophysiology data was acquired at $30\,\mathrm{kHz}$ using a $11\,\mathrm{\mu m} \times 15\,\mathrm{\mu m}$ multi-shank linear silicon probe (128 channels) implanted in the center of the left monocular V1. One wide-angled camera (around $120°$) was aimed outwards to capture the visual scene available to the right eye of the mouse at $16\,\mathrm{ms}$ per frame ("worldcam"). A second camera was aimed at the right eye (illuminated by an infrared-LED) to record a video feed of the right eye at $30\,\mathrm{Hz}$. The IMU acquired three-axis gyroscope and accelerometer data at $30\,\mathrm{kHz}$. In addition, a top-down camera recorded the mouse in the arena at $60\,\mathrm{Hz}$.

During experiments, mice were placed in an arena where they could move around freely for about 1 hour. The arena was approximately $48\,\mathrm{cm}$ long $\times$ $37\,\mathrm{cm}$ wide $\times$ $30\,\mathrm{cm}$ high. The gray floor was covered with black-and-white Legos to provide visual contrast. One wall of the arena was a monitor displaying a moving black-and-white spots stimulus, and the other three walls were covered with wallpaper with static stimuli including white noise, black-and-white high-spatial-frequency gratings, and black-and-white low-spatial-frequency gratings. In order to encourage foraging behavior during the recording sessions, small fragments of tortilla chips were sparsely distributed across the arena.

The worldcam video was downsampled to $60 \times 80$ pixels. DeepLabCut [43] was used to extract extract eye position ($\theta$, $\phi$) and pupil radius ($\sigma$). Pitch ($\rho$) and roll ($\omega$) of the mouse's head position were extracted from the IMU. Locomotion speed ($s$) was estimated from the top-down camera feed using DeepLabCut [43]. Electrophysiology data bandpass-filtered between $0.01\,\mathrm{Hz}$ and $7.5\,\mathrm{kHz}$, and spike-sorted with Kilosort 2.5 [44]. Single units were selected using Phy2 (see [45]) and inactive units (mean firing rate $< 3\,\mathrm{Hz}$) were removed. This yielded 68, 32, and 49 active units for Mouse 1–3, respectively. To prepare the data for machine learning, all data streams were deinterlaced and resampled at $20.83\,\mathrm{Hz}$ ($48\,\mathrm{ms}$ per frame; Fig. 1C).

# B    Vision-Only Models

## B.1    Hyperparameter Tuning

We performed a grid search to find the optimal CNN kernel size (3, 5, 7, 9), number of channels (32, 64, 128, 256, 512; in various combinations), and dropout rate (0, 0.25, 0.5). While other models often rely on kernel size 3 for their CNN, we found these small kernels to lead to worse performance, perhaps due to the mouse's low-resolution vision. Kernel size 7 performed best.

We repeated the grid search for CNNs with different number of convolutional layers. The resulting 3-layer CNN with 0.5 dropout rate outperformed many differently sized networks, such as a 1-layer CNN with 1024 channels (i.e., a shallow but wide network), a 2-layer CNN, or a 4-layer CNN. Choice of learning rates and optimizers had no notable effect on the final performance of the networks.

## B.2    Autoencoder

We initially hypothesized that an autoencoder could provide regularization benefits over a "vanilla" CNN, because the reconstruction loss might encourage the model to learn visual features that are useful for decoding. We used an encoder $\phi$ to map the original frame $\mathcal{F}$ to a vector $\mathcal{V}$ in the latent space, which was present at the bottleneck, while the decoder $\psi$ then mapped the vector $\mathcal{V}$ from the latent space to the output.

$$\phi : \ \mathcal{F} \to \mathcal{V}, \tag{1}$$

$$\psi : \ \mathcal{V} \to \mathcal{F}, \tag{2}$$

$$\phi, \psi = \mathrm{argmin}_{\phi,\psi} ||\mathcal{F} - (\psi \cdot \phi)\mathcal{F}||^2. \tag{3}$$

After hyperparameter search, we settled on size 256 for the latent space vector, and the weight of the reconstruction loss relative to the Poisson loss was fixed at 0.5. Both the encoder and the decoder were 3-layer CNNs, and their numbers of channels were symmetric. However, after testing a number of autoencoders with different configurations (Table 4), we found that a simple 3-layer CNN outperformed any and all of them.

| Kernel size, encoder #channels | Mouse 1 | | Mouse 2 | | Mouse 3 | |
|---|---|---|---|---|---|---|
| | $cc \uparrow$ | MSE $\downarrow$ | $cc \uparrow$ | MSE $\downarrow$ | $cc \uparrow$ | MSE $\downarrow$ |
| 3, $16 \times 32 \times 64$ | $.539 \pm .149$ | .0728 | $.389 \pm .128$ | **.107** | $.502 \pm .129$ | .0996 |
| 5, $16 \times 32 \times 64$ | $.550 \pm .147$ | .0728 | $.363 \pm .116$ | .109 | $.508 \pm .135$ | .0983 |
| 7, $16 \times 32 \times 64$ | $.525 \pm .152$ | .0732 | $.353 \pm .121$ | .117 | $.509 \pm .131$ | **.0980** |
| 9, $16 \times 32 \times 64$ | $.518 \pm .147$ | .0752 | $.315 \pm .101$ | .119 | $.492 \pm .135$ | .0997 |
| 3, $32 \times 64 \times 128$ | $.543 \pm .144$ | .0737 | $.367 \pm .128$ | .109 | $.503 \pm .131$ | .100 |
| 5, $32 \times 64 \times 128$ | $.551 \pm .149$ | .0723 | $.361 \pm .109$ | .119 | $.514 \pm .132$ | .0984 |
| 7, $32 \times 64 \times 128$ | $.539 \pm .145$ | .0739 | **.390 $\pm$ .118** | .109 | $.492 \pm .129$ | .100 |
| 9, $32 \times 64 \times 128$ | $.510 \pm .155$ | .0758 | $.331 \pm .119$ | .112 | $.500 \pm .134$ | .101 |
| 3, $64 \times 128 \times 256$ | $.541 \pm .146$ | .0758 | $.374 \pm .123$ | .110 | $.514 \pm .127$ | .0990 |
| 5, $64 \times 128 \times 256$ | $.552 \pm .145$ | .0777 | $.362 \pm .119$ | .110 | $.508 \pm .134$ | .104 |
| 7, $64 \times 128 \times 256$ | **.553 $\pm$ .134** | **.0688** | $.369 \pm .104$ | .111 | **.530 $\pm$ .136** | .0992 |
| 9, $64 \times 128 \times 256$ | $.537 \pm .146$ | .0811 | $.355 \pm .109$ | .119 | $.500 \pm .128$ | .105 |

Table 4: Performance of different autoencoders. The numbers of channels in the decoder were symmetric with those of the encoder. Best performing networks are indicated in bold. $cc$: cross-correlation, mean $\pm$ standard deviation across neurons ($\uparrow$: the higher the better), MSE: mean-squared error ($\downarrow$: the lower the better).

## C   Visual Receptive Fields

To test whether the recovered visual receptive fields are sensitive to the choice of initial values for the behavioral variables, we repeated our experiments by initializing the behavioral variables with noise sampled from the uniform distribution $[-1, 1)$. The values remained the same throughout the process of gradient ascent.

We found that different values of behavioral variables resulted in similar visual receptive fields. A representative example is shown in Fig. C.1. Although some of the absolute values changed, receptive field structure stayed qualitatively the same, with identical excitatory and inhibitory subregions to the ones reported in Fig. 4.

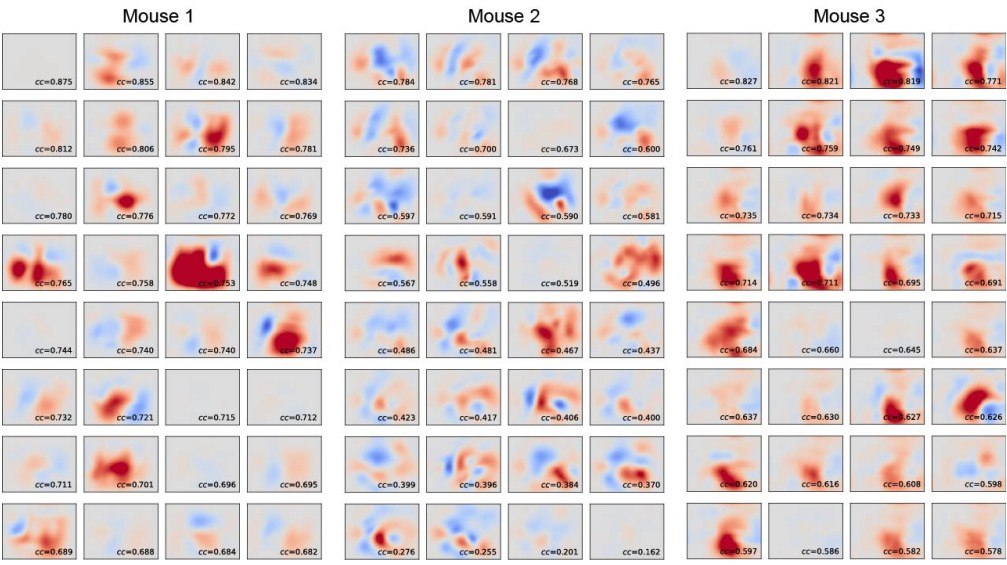

Figure C.1: The maximally activating stimuli learned in CNN+GRU$_1$, generated via gradient ascent with a fixed and randomly initialized behavioral variable. The 32 neurons with the highest cross-correlation ($cc$) from each mouse are shown, sorted by $cc$.

