# OpenReview forum: "Multimodal Deep Learning Model Unveils Behavioral Dynamics of V1 Activity in Freely Moving Mice"
_NeurIPS.cc/2023/Conference — NeurIPS 2023 poster_

### Official Review · Reviewer_xdTa · 2023-06-28

**Soundness:** 4 excellent
**Presentation:** 4 excellent
**Contribution:** 4 excellent
**Rating:** 8
**Confidence:** 3

**Summary:**

The authors propose a neural network for predicting activity of V1 neurons in freely moving mice. The data consists of ~hour long electrophysiological recordings of a neuronal population while the mouse is freely exploring the space. Simultaneously with the neural activity, the experimental setup allows the authors to record the visual scenes from the mouse perspective as well as a number of behavioral variables.

The network consists of two modules: one computes visual features, the other encodes behavioral variables. The outputs of both modules are concatenated and fed into the recurrent unit (GRU) accounting for the temporal dynamics, the output of which is decoded into the predicted neural activity of each neuron in the recorded population.

The models achieve state-of-the-art performance both among the visual models and among those including behavioral variables. The authors analyze the model by computing MEIs as well as performing saliency analysis to study the effect of individual behavioral variables.

**Strengths:**

I think it is a very good paper.

+ It is clearly written
+ It discusses an important problem (integration of behavioral variables in predicting mouse V1 activity)
+ The proposed model achieves a noticeable improvement over the previous state-of-the-art

**Weaknesses:**

I don't see any significant methodological weaknesses.

I think the main limitation is the small size of the dataset (a few dozen neurons per mouse). For example, the diversity of the MEIs in Fig. 4 both within and across the animals is very interesting, but a fairly small dataset size doesn't allow us to draw any quantitative conclusions about these MEIs beyond visual inspection. It would be really nice to have a calcium dataset of thousands of neurons in a freely moving mouse (similar to the Sensorium one) to address this issue, but it is of course not a critique of this paper.

**Questions:**

- I am surprised to see that your vision-only network quite significantly outperforms the baselines despite having a standard CNN-FC architecture. Why do you think it is the case? And specifically for mouse 2, why do you think your model is able to perform so much better than the baselines?

- Do you think the training data could leak into the test set in the 70-30 splits of continuous segments as you described in the paper? I guess it is not unlikely that a freely moving mouse could be exploring the same part of the arena in some of the train and test segments?

- How did you set the values of the behavioral variables for the visual MEIs computation? It would be very interesting to see how the MEIs change as a function of behavioral variables as the animal explores the space around it.

- Some of the previous work (e.g. [11]) explicitly assumed the existence of localized receptive fields by factorising the readout into sparse spatial masks and a feature vector. But looking at the MEIs you generated, many neurons don't have receptive fields, so such an architecture doesn't seem reasonable. Do you have any ideas why the models with an explicit receptive field assumption perform well nevertheless?

**Limitations:**

The limitations are adequately addressed in the paper.

---

> ### Author Rebuttal · Authors · 2023-08-09
>
> We thank the reviewer for their encouraging comments.
>
> > It would be really nice to have a calcium dataset of thousands of neurons in a freely moving mouse (similar to the Sensorium one)
>
> We agree with the reviewer that creating a large and standardized neural recording dataset from freely moving mice would be the dream. We outlined this in our future work section. Unfortunately, to date, Calcium imaging in a freely moving animal is not feasible. (Calcium imaging was possible in the Sensorium competition because the animals were head-fixed.) A yield of ~100 units is considered state of the art for freely moving animals. Furthermore, it is not clear whether the fast neuronal dynamics associated with locomotion could be captured with Calcium imaging, which is relatively slow.
>
> > I am surprised to see that your vision-only network quite significantly outperforms the baselines despite having a standard CNN-FC architecture.
>
> We were also surprised to see that vanilla CNNs were better than the other architectures we experimented with, including autoencoders, variational autoencoders, filter bank models, and pre-trained deep neural networks (ResNet and EfficientNet). In our opinion, the reason might be that the neurons in the mouse visual cortex were selective for other behavioral inputs and the vanilla CNN architecture imposed the fewest assumptions about how the visual input contributed to the neural activity. This is especially true for Mouse 2: we learned that the cortical probes of Mouse 2 were more superficial compared to the other two mice, so the recorded neurons may have both different anatomical inputs and different visual responses.
>
> > Do you think the training data could leak into the test set in the 70-30 splits of continuous segments as you described in the paper? I guess it is not unlikely that a freely moving mouse could be exploring the same part of the arena in some of the train and test segments?
>
> While it is possible that the mouse could have been exploring the same part of the arena at different segments of the recording, it was free to move its head, eyes, and body as it saw fit. Thus “data leak” is unlikely as two duplicate data points could only be produced by the animal exactly duplicating the time courses of its eye, head, and body movement in the exact same location of the arena.
>
> > How did you set the values of the behavioral variables for the visual MEIs computation? It would be very interesting to see how the MEIs change as a function of behavioral variables as the animal explores the space around it.
>
> The behavioral variables were initialized to a vector of all ones, and updated in the loop with the visual MEI. We agree that setting a fixed behavioral variable vector and inspecting the changes in visual MEIs would be an interesting analysis, which we would be happy to provide in the camera-ready version.
>
> > Some of the previous work (e.g. [11]) explicitly assumed the existence of localized receptive fields by factorising the readout into sparse spatial masks and a feature vector. [...] Do you have any ideas why the models with an explicit receptive field assumption perform well nevertheless?
>
> We agree with the reviewer that enforcing a localized receptive field may hinder good model fits. In contrast to previous work, our analysis used gradient ascent to reveal the maximally exciting visual input. We speculate that previous models may have still worked well because the assumption of a localized receptive field was imposed on the final feature map output (e.g., [11]), which is drastically different (and at a coarser resolution) than the raw visual input. Therefore, the receptive field in those models was not exactly as localized as claimed, which may explain why these models still performed well.

---

> > ### Comment · Reviewer_xdTa · 2023-08-10
> >
> > Thank you very much for your response and for addressing my questions! I don't have any further questions at this point.

---

### Official Review · Reviewer_riS4 · 2023-07-04

**Soundness:** 3 good
**Presentation:** 3 good
**Contribution:** 2 fair
**Rating:** 5
**Confidence:** 5

**Summary:**

The authors use convolutional neural networks to fit data from neuronal recordings in primary visual cortex (V1) of the mouse while the animal is freely exploring the environment. The model incorporates visual signals but also other behavioral variables (“multimodal” aspect). The proposed model provides better fits to neuronal data than other benchmark models. The neuronal and behavioral data are particularly interesting and can provide interesting insights into neuronal computations when interpreted with mechanistic models.

**Strengths:**

The ability to fit neural data in a dynamic fashion is interesting and distinguishes this work from previous work in the field.

Numerically, the proposed model and incorporation of behavioral variables provide better fits to the data than the alternatives tested.
The differences across different neurons (section starting on line 212) is quite interesting, especially if it can be connected in the future to the ongoing efforts to characterize different cell types in mice.

The neuronal and behavioral data are particularly interesting and probably the highlight of the paper. These data can provide interesting insights into neuronal computations when interpreted with mechanistic models and would be very useful for future investigators that are interested in building theories and models of brain function.

**Weaknesses:**

One could imagine incorporating a lot of different world variables to fit neuronal responses. Here the authors choose a very reasonable set of such variables, including pupil size, head direction, moving speed. What exactly this means in terms of the function of V1 is not clear.

This kind of neural data fitting has become extremely common in the field. However, it remains unclear what kind of conclusions one can draw from this type of data fitting.
Take the first sentence of the results. Table 1 does show that the 3-layer CNN is slightly better than other models, as the subtitle indicates. Then what? Does this mean that mouse V1 functions like a 3-layer CNN? Clearly not. There are lots of problems with 3-layer CNNs, including adversarial attacks, lack of robustness to noise, etc. These issues are not tested here. Thus, it is difficult to state what we learn about V1 function from the fact that the 3-layer CNN is slightly better than, say, ResNet-18. It is certainly better in fitting data from this experiment, what this means in the big picture of V1 computations is unclear.

The next section indicates that behavioral variables improve the data fitting, which I agree with. However, it remains very unclear that V1 neurons would have inputs that directly reflect the behavioral variables incorporated into the model. Perhaps V1 neurons could get indirect inputs that relate to pupil size, head direction, running speed, etc. From a neuroscience perspective, the critical question is to understand the anatomical inputs, the mechanisms underlying these dependences.

The notion of mixed selectivity alluded to in the discussion has been extensively used in neuroscience and remains rather useless. Neurons show “mixed selectivity” only insofar as the investigators use variables that are not directly related to the neuronal responses. Without mechanistic models, there are lots of representations that might seem as “mixed selectivity”. As a trivial example, consider a neuron that responds to 45 degrees orientation preferentially. But the investigators try to fit the responses in terms of horizontal and vertical and then claim “mixed selectivity!”. There is obviously no such thing, the neuron represents 45 degrees, period . The claims in the paper follow the same logic.



**Questions:**

The differences across different neurons (section starting on line 212) is quite interesting, especially if it can be connected in the future to the ongoing efforts to characterize different cell types in mice. Expanding on this section would be useful in terms of neuroscience. What is distinct about each type of neuron, how do their properties relate to their firing rates, variability, receptive field sizes and properties, etc. This is where the paper can go from data fitting into first steps into a mechanistic understanding of V1 properties.

**Limitations:**

There is no discussion about limitations, of which there are plenty.

---

> ### Author Rebuttal · Authors · 2023-08-09
>
> We thank the reviewer for their detailed feedback and constructive comments. We agree that data fitting is not the final answer to understanding the functioning of mouse visual cortex, and we do not presume to be able to provide such an answer. However, we believe that our modeling efforts may provide valuable insights about visual cortex function, potentially leading to mechanistic insights in the future. For instance, previous models of mouse V1 have seen limited success in incorporating behavioral variables in their prediction (e.g., Sensorium+ competition). Rather than concluding that behavioral input may play a secondary role in visual cortex function, our study suggests that the move to freely-moving datasets is paramount to revealing the role of behavioral input to predictions of V1 activity. Moreover, identifying the behavioral variables that modulate processing (and particularly their impact on coding) may guide future experimental efforts to identify their neural basis.
>
> > However, it remains very unclear that V1 neurons would have inputs that directly reflect the behavioral variables incorporated into the model. Perhaps V1 neurons could get indirect inputs that relate to pupil size, head direction, running speed, etc. From a neuroscience perspective, the critical question is to understand the anatomical inputs, the mechanisms underlying these dependences.
>
> We agree with the reviewer in general. However, our results are indeed aligned with recent neuroscientific work that highlights how locomotion, arousal, and motor signals enter V1 (Fu et al., 2014, doi:10.1016/j.cell.2014.01.050; Leinweber et al., 2017, doi:10.1016/j.neuron.2017.08.036). Froudarakis et al. 2019 (doi:10.1146/annurev-vision-091517-034407) and Parker et al. 2020 (doi:10.1016/j.tins.2020.05.005) provide a thorough review of non-visual inputs to visual cortex. Given that these anatomical inputs exist, the question remains how V1 neurons may integrate these behavioral variables with visual information. To this end, our study provides a computational account of how V1 neurons may combine these multimodal inputs during free exploration.
>
> > The notion of mixed selectivity alluded to in the discussion has been extensively used in neuroscience and remains rather useless.
>
> We agree with the reviewer that “mixed selectivity” should be clearly defined in order to be meaningful. In the context of this study, mixed selectivity refers to a neuron encoding two different categories of information: e.g., visual and motor signals, or stimulus and reward signals. Combining horizontal and vertical edges would not count as such; this would be a purely visual response. We will make sure to clearly define this term in the camera-ready version.
>
> > The differences across different neurons (section starting on line 212) is quite interesting, especially if it can be connected in the future to the ongoing efforts to characterize different cell types in mice. Expanding on this section would be useful in terms of neuroscience.
>
> We agree with the reviewer that an important next step is to connect our results to characterizing different cell types in mice from a mechanistic and anatomical point of view. While at present we can speculate about these differences based on our model fits, to thoroughly answer this question, we would require a different set of tools that are beyond the scope of this work.

---

> > ### Comment · Reviewer_riS4 · 2023-08-10
> > **Reasonable responses, still unconvinced**
> >
> > The authors provide very reasonable responses.
> > Indeed, there are plenty of non-visual inputs to V1 in mice. Then what? Do those inputs help the mouse see better? Are there particular behaviors that are contingent on those connections? Of note, the current work does not present a mechanistic model of what those connections do. All we can say is that adding some behavioral variables can lead to better fitting of V1 activity. For example, it could be that mice pay more attention when they are running and that the neurons have higher activity and that leads to better correlation coefficients, but this should not be confounded with a mechanistic understanding of the function of V1 neurons.
> >
> > Most neurons have many inputs, sometimes as many as 10,000 inputs. Perhaps what the authors mean by mixed selectivity is that there are local and non-local anatomical connections and that there are many neurons that have different non-local anatomical connections.
> >
> > That said, there is nothing technically wrong with this study, as far as I can tell. While we can disagree on whether this study advances neuroscience understanding in any way, the work is well done, clear and well written.

---

### Official Review · Reviewer_NhDi · 2023-07-07

**Soundness:** 3 good
**Presentation:** 4 excellent
**Contribution:** 3 good
**Rating:** 7
**Confidence:** 3

**Summary:**

The authors introduce a multimodal recurrent neural network to integrates information beyond vision - behavioral and temporal dynamics processed by a separate head to explain V1 activity in mice. This model (vision + behavioral and temporal dynamics) is the state-of-the-art in prediction of V1 activity during free exploration. Further, this model is analyzed using maximal-activating stimuli and saliency maps, to obtain insights into function of mice V1 area.

**Strengths:**

* The paper is well written and the integration of information beyond vision (or any one modality) for neural predictivity is novel, as far as I know.

* The results are strong and convincing with proper ablation studies

**Weaknesses:**

* Vision only model seems to have fewer parameters compared to the ones with GRU on top. This makes it hard to say if the improvements come from the extra parameters (unlikely).

**Questions:**

* I am surprised ResNet models do not do as well the smaller CNN. The skip connections in the resnet should have resulted in the model being at least as good as the shallower model. Have you looked into why?

---

> ### Author Rebuttal · Authors · 2023-08-09
>
> We thank the reviewer for their encouraging comments and suggestions.
>
> > Vision only model seems to have fewer parameters compared to the ones with GRU on top. This makes it hard to say if the improvements come from the extra parameters (unlikely).
>
> It is true that our vision-only models had fewer parameters than our vision-and-behavior models - this was unavoidable as adding behavioral and temporal dynamics increased the input space and thus necessitated more parameters. However, we agree with the reviewer that the performance improvements were not primarily due to the added number of parameters, for several reasons. First, we found that adding more parameters to the vision-only model did not improve performance. For this, we investigated several vision-only alternatives, which are described in the appendix, such as adding additional layers (Appendix A.1), more channels (Appendix A.1), and a decoder architecture  (Appendix A.2 and Table 4). Second, we experimented with both GRU and LSTM architectures, and found that GRU (despite the lower number of extra parameters) drastically outperformed LSTM.
>
> > I am surprised ResNet models do not do as well the smaller CNN. The skip connections in the resnet should have resulted in the model being at least as good as the shallower model. Have you looked into why?
>
> The reviewer raises a great point. During our early experiments with different ResNet architectures, we noticed that these models were drastically overfitting our dataset. It is possible that this problem could be alleviated with more data, but as is, our dataset is already large by experimental standards in the field. Another possibility is that mouse visual cortex has a shallower architecture than macaque visual cortex (as pointed out in the main text), thus making deeper ResNet architectures less suitable for the task.

---

> > ### Comment · Reviewer_NhDi · 2023-08-16
> > **Reply to authors**
> >
> > Thank you for the response.
> >
> > A difference of this work with [1] and others is that in their case, the models are initially optimized for a task (usually image categorization) and then the features from the model is used to predict neural data. Whereas, in this work, the authors directly optimize the entire network to predict the neural data. I am curious to hear as to why and if the authors expect to see any difference if they were to do it with a performance-optimized to network?
> >
> > In any case, I think the work is interesting & I am more convinced than before, thus I am improving my initial score.
> >
> > [1] Daniel L. K. Yamins, Ha Hong, Charles F. Cadieu, Ethan A. Solomon, Darren Seibert, and James J. DiCarlo. Performance-optimized hierarchical models predict neural responses in higher visual cortex. Proceedings of the National Academy of Sciences, 111(23):8619–8624, June 2014. Publisher: Proceedings of the National Academy of Sciences.

---

> > > ### Author Response · Authors · 2023-08-18
> > >
> > > We thank the reviewer for their encouraging comments. Our ResNet-18 model and EfficientNet-B0 model were finetuned from the weights pre-trained for the image classification task, but in the end, they did not result in superior performance compared to vanilla CNNs (Table 1). We believe that this is due to the unique properties of the mouse visual system which we reviewed in lines 51-63. A key difference between our work and [1] is the type of neural data. The models in [1] were used to fit neural data from macaques, but our work tries to predict neural activity from mice. In contrast to primates, mice are known to have a low-resolution vision and a shallower, more parallel, and more interconnected visual cortex, which is not a close match to the usually deeper, more feed-forward performance-optimized networks. In addition, mice’s visual processing is more related to movement instead of a passive analysis of the visual scene, and a large proportion of examples from image-categorization datasets may not match what a mouse sees in its surroundings. This is another reason why the models pre-trained with an image classification objective do not perform well in this case.

---

> > > > ### Comment · Reviewer_NhDi · 2023-08-18
> > > >
> > > > Thank you. That is not what I meant, let me clarify my question. I do not mean using ImageNet optimized networks rather a model trained for some task on the _data you collected_. This could be scene segmentation, categorization or even self-supervised training if you want to avoid annotating your data (something like autoencoder-like training). Then you freeze the network and only train a linear readout to predict neural responses. This would make it more akin to [1] in that the feature extractors are performance optimized and you are probing to see how well these features can predict neural data. In any case, this is sort of orthogonal to this work, I am just curious to hear your opinion.

---

> > > > > ### Author Response · Authors · 2023-08-20
> > > > >
> > > > > Thank you for your clarification. We expect that if the model is optimized for a task ethologically relevant to mice, then the features extracted by the model might predict the neural data relatively well. During our early experiments, we did try a version where the pre-trained ResNet model was treated as a fixed feature extractor: The ResNet weights were frozen except that the last linear layer was replaced and retrained. It did not work well, and it is to be expected because image classification is not relevant to the mouse. However, we speculate that if the task is somehow related to the characteristics of visual processing in mice, then the model could potentially extract useful features and thus perform better.

---

### Official Review · Reviewer_ZQ1m · 2023-07-08

**Soundness:** 3 good
**Presentation:** 2 fair
**Contribution:** 3 good
**Rating:** 6
**Confidence:** 4

**Summary:**

The authors in this work propose a novel multimodal approach to design encoder models of mouse visual processing. The authors identify the limitations of unnaturalistic (head-fixed) recording, limited behavioral inputs and lack of temporal dynamics in prior recordings and models of mouse visual system and address these with new data recorded from 3 freely-moving mice that they have access to which fixes the above-mentioned issues. The multimodal architecture, a CNN-GRU encoder simultaneously encodes time-locked visual input and elaborate mouse behavioral state (running speed, eye and head position, pupil size, the first derivatives and pairwise products of these quantities). A neural response prediction readout is attached to the output of the encoder. The authors show that a 3-layer CNN is the best predictor of mouse visual responses and that behavioral input consistently outperforms a purely vision-based encoder. Further gradient ascent visualization and saliency-based analyses add interpretability to the above encoder.

**Strengths:**

+ The proposed work is quite original and sound in the joint-computational modeling of mouse visual and behavioral responses in freely-moving mice. The encoder architecture proposed by the authors is quite interpretable and empirically performs well to predict the neural activity data. The originality in this work in my opinion is from the modeling that the authors propose. The dataset that has been used here seems to not be original to this work but prior work referenced by the authors in the Methods section.
+ It is (although not surprising) clear from the experimental results that joint-encoding of vision and behavior outperforms predictions from a purely vision based model. The methods and results sections have been written quite well in adequate details on preprocessing techniques, training of the encoder and metrics used to evaluate the quality of visual encoding.
+ Additional visualization and saliency based analysis performed by the authors adds interpretability to the proposed encoder model as it shows well-defined receptive fields (mostly those model neurons with high correlation to Mouse 1 neurons though) and selectivity to various behavioral attributes.

**Weaknesses:**

- Unfortunately it seems that the dataset has been published already and is accessed by this submission; the modeling efforts built on top seem like a relatively small contribution in my opinion. Especially since the kind of architecture designed and its hyperparameters aren't justified with a systematic search. Did the authors attempt alternative architectural choices, normalization techniques other than BatchNorm, different training approaches (unsupervised / self-supervised) or other recurrent units that are not included in this submission? Please provide the required justification for the presented architectural choices.
- Related work section seems to be quite limited and I believe it could be enhanced to further include more information about other relevant computational models of mouse visual activity.
- Other than Figure 1 and Figure 4, the other figures seem to have poor spatial resolution and need to be improved in quality for better readability.
- Very little information is included about the neural activity dataset, I believe it would be good to add more details about the kind of visual stimuli used and about the recording setup in the Methods section or in the supplementary information.
- I could not find mention of whether the authors are open to sharing their models and code, this would be appreciated if the authors want to maximize the reproducibility of their work.

**Questions:**

Please find my suggestions in the weaknesses section of my review.

**Limitations:**

Limitations have been adequately addressed in this submission.

---

> ### Author Rebuttal · Authors · 2023-08-09
>
> We thank the reviewer for their encouraging comments and are pleased that they agree about the originality of the work. We do believe that this work constitutes a significant modeling contribution, as most previous modeling attempts have focused on head-fixed datasets with static stimuli, which may be appropriate for foveating animals, but arguably provide limited insight into visual processing in real-world environments. The extension to freely-moving datasets is not trivial, and the present work demonstrates state-of-the-art performance on a dataset collected with state-of-the-art equipment.
>
> > Please provide the required justification for the presented architectural choices. Did the authors attempt alternative architectural choices, normalization techniques other than BatchNorm, different training approaches (unsupervised / self-supervised) or other recurrent units that are not included in this submission?
>
> We indeed experimented with several different network architectures (inspired by the results of the Sensorium competition) prior to arriving at the proposed network. This included autoencoders (whose details are described in the appendix), variational autoencoders, pre-trained deep neural networks (ResNet and EfficientNet), filter bank models, and LSTM/RNN networks. The hyperparameter search for the CNN architecture is described in detail in the appendix.
> We did not attempt normalization techniques other than BatchNorm, because we found BatchNorm to be most effective in our previous experience with multimodal modeling and BatchNorm was standard in the field [11], [12]. We did not try unsupervised or self-supervised learning because our task (predicting the firing rate of each neuron based on visual and behavioral inputs) was supervised in nature and it was not clear how to formalize it so that other training techniques could be applied. We agree this could be a promising avenue for future research.
>
> > Related work section seems to be quite limited and I believe it could be enhanced to further include more information about other relevant computational models of mouse visual activity
>
> We agree that this section should be expanded to include other relevant computational models of the mouse visual activity. We will make sure of that in the camera-ready version.
>
> > Other than Figure 1 and Figure 4, the other figures seem to have poor spatial resolution and need to be improved in quality for better readability.
>
> Thank you for pointing out the image resolution issue. We will ensure the figures are of high spatial resolution in the camera-ready version.
>
> > Very little information is included about the neural activity dataset
>
> We agree that more information about the neural activity dataset could facilitate the flow of the paper, but we did not go into too much detail because space was limited and because it was reported in detail in the original publication which described the experimental results and the dataset that we accessed [24]. We will make sure to include a more detailed description of the dataset in the appendix in the camera-ready version.
>
> > I could not find mention of whether the authors are open to sharing their models and code
>
> The reviewer raises an important point, and we are happy to share all pre-trained models and code (line 151).

---

### Author Rebuttal · Authors · 2023-08-09

We thank the reviewers for their constructive comments. We are pleased that reviewers agreed this is a clearly written paper (ZQ1m, xdTa) describing novel work (ZQ1m, NhDi) with strong results (NhDi, riS4, xdTa) that may help build theories and models of brain function (riS4).
Reviewers raised insightful questions about the details of the dataset (ZQ1m, xdTa), the choice of our model architecture and comparison to alternative architectures (ZQ1m, NhDi, xdTa), and the implications for our mechanistic understanding of V1 processing (riS4).

Below we address each of the reviewer’s questions and concerns point by point. In short, the dataset we accessed in this work is the state-of-the-art setup for collecting neural activity data from freely moving mice. We have indeed performed an architecture search to arrive at our novel multimodal network architecture (see Appendix). While our selection of behavioral variables was largely guided by known computational properties of mouse V1, our multimodal model may help unveil the computational principles by which non-visual inputs are integrated in mouse visual cortex. Lastly, we would like to emphasize that we are happy to provide all our pre-trained models and code upon acceptance.

---

### Decision · Program_Chairs · 2023-09-21

**Decision:**

Accept (poster)

**Comment:**

The reviewers agree that it’s a technically solid paper with a novel contribution that advances the field of predictive modeling of neural responses. Some reviewers raised the concern that the paper’s contribution may be “just a data fitting exercise” and questioned whether such “data fitting” is useful. While this (somewhat philosophical) discussion is certainly both interesting and important, it applies to the entire field of predictive modeling and is probably better kept separately from the decision about this individual paper.